# Assessment of the Contribution of a Thermodynamic and Mechanical Destabilization of Myosin-Binding Protein C Domain C2 to the Pathomechanism of Hypertrophic Cardiomyopathy-Causing Double Mutation *MYBPC3*^Δ*25bp/D389V*^

**DOI:** 10.3390/ijms222111949

**Published:** 2021-11-04

**Authors:** Frederic V. Schwäbe, Emanuel K. Peter, Manuel H. Taft, Dietmar J. Manstein

**Affiliations:** 1Fritz Hartmann Centre for Medical Research, Institute for Biophysical Chemistry, Hannover Medical School, Carl Neuberg Str. 1, D-30625 Hannover, Germany; Schwaebe.Frederic@mh-hannover.de (F.V.S.); Peter.Emanuel@mh-hannover.de (E.K.P.); Taft.Manuel@mh-hannover.de (M.H.T.); 2Division for Structural Biochemistry, Hannover Medical School, Carl Neuberg Str. 1, D-30625 Hannover, Germany

**Keywords:** hypertrophic cardiomyopathy, heart disorder, MyBPC, enhanced molecular dynamics simulations, allosteric trigger, cardiac contractility, force generation, protein unfolding

## Abstract

Mutations in the gene encoding cardiac myosin-binding protein-C (MyBPC), a thick filament assembly protein that stabilizes sarcomeric structure and regulates cardiac function, are a common cause for the development of hypertrophic cardiomyopathy. About 10% of carriers of the Δ25bp variant of *MYBPC3*, which is common in individuals from South Asia, are also carriers of the D389V variant on the same allele. Compared with noncarriers and those with *MYBPC3*^Δ*25bp*^ alone, indicators for the development of hypertrophic cardiomyopathy occur with increased frequency in *MYBPC3*^Δ*25bp/D389V*^ carriers. Residue D389 lies in the IgI-like C2 domain that is part of the N-terminal region of MyBPC. To probe the effects of mutation D389V on structure, thermostability, and protein–protein interactions, we produced and characterized wild-type and mutant constructs corresponding to the isolated 10 kDa C2 domain and a 52 kDa N-terminal fragment that includes subdomains C0 to C2. Our results show marked reductions in the melting temperatures of D389V mutant constructs. Interactions of construct C0–C2 D389V with the cardiac isoforms of myosin-2 and actin remain unchanged. Molecular dynamics simulations reveal changes in the stiffness and conformer dynamics of domain C2 caused by mutation D389V. Our results suggest a pathomechanism for the development of HCM based on the toxic buildup of misfolded protein in young *MYBPC3*^Δ*25bp/D389V*^ carriers that is supplanted and enhanced by C-zone haploinsufficiency at older ages.

## 1. Introduction

Hypertrophic cardiomyopathy (HCM) is the most common genetic cardiac disorder [1,2,3,4]. It is mainly characterized by an increased left ventricular wall thickness and myocyte disarray [5]. Clinical effects vary widely, ranging from no symptoms to shortness of breath, chest pain, atrial fibrillation, or sudden cardiac death. This is related to the fact that the disease can be caused by more than 1400 mutations in at least 11 different genes, with *MYBPC3* and *MYH7* alone responsible for about 70–80% of cases [2,6,7].

Mutations in sarcomeric genes often perturb the carefully regulated interplay of thick myosin-containing and thin actin-containing filaments, which is essential for cardiac function. During contraction, myosin periodically interacts with actin in a Ca^2+^-dependent manner, with tropomyosin and the troponin complex playing a regulatory role. In addition, cardiac myosin-binding protein C (MyBPC) carries out multiple regulatory functions. Cardiac MyBPC (encoded by the gene *MYBPC3*) is one of three MyBPC isoforms, alongside its slow (*MYBPC1*) and fast (*MYBPC2*) counterparts in skeletal muscle. MyBPC is a modular protein of 140 kDa that comprises eight IgI-like domains, three FnIII-like domains, and one unique M domain (Figure 1A). The M domain harbors multiple phosphorylation sites that modulate protein–protein interactions and are implicated in cardiac disease (as reviewed by [8,9]). MyBPC is anchored to the thick filament via its C-terminal C8–C10 domains [10,11,12], while its N-terminal domains interact with actin and different myosin subdomains [13,14,15,16,17,18,19,20]. Binding to myosin S2 has been demonstrated to promote myosin’s super-relaxed state, limiting the number of heads accessible for contraction [21,22,23]. Its interaction with cardiac thin filaments is thought to sensitize the sarcomere for Ca^2+^ by azimuthally displacing tropomyosin [24,25]. Moreover, it can apply a viscous load by tethering the thick and the thin filament, thus acting as a brake to cardiac contraction with precisely defined mechanochemical properties [26,27,28].

The majority of pathogenic *MYBPC3* mutations result in the production of a truncated protein, inducing HCM through a reduced amount of functional MyBPC (haploinsufficiency) [32,33,34,35,36,37]. In contrast, single amino acid replacements often destabilize a subdomain or alter protein–protein interactions, promoting development of HCM via alternative pathomechanisms [32,38,39,40,41,42].

The common *MYBPC3*^Δ*25bp*^ variant with a 25 base pair deletion in intron 32 has been associated with an increased odds ratio of 6.99 for the development of HCM, affecting an estimated 100 million carriers in the South Asian population [43,44,45]. These studies show that transcripts originating from the affected allele are subject to alternative splicing mechanisms that lead to skipping of exon 33. The resulting protein, which has an altered C10 domain with 62 residues removed and 55 newly added, mislocalizes to the Z-disc. It has been shown that overproduction of the modified protein is sufficient to cause HCM in transgenic mice [46,47]. However, whether a corresponding alternative splicing process in humans also results in the production of significant amounts of the modified protein and its mislocalization remains to be confirmed.

More recently, the D389V mutation in the C2 domain was discovered in ~10% of the *MYBPC3*^Δ*25bp*^ carriers, constituting the monoallelic *MYBPC3*^Δ*25bp/D389V*^ double mutation [48]. Individuals carrying this mutation exhibit increased left ventricular ejection fraction, left ventricular fractional shortening, and cardiomyocyte hypertrophy. These features, which are considered early findings of HCM, are not observed in an *MYBPC3*^Δ*25bp*^ control cohort.

While many previous studies utilized C0–C2 or C1–C2 constructs [13,14,15,16,17,18,19,20], little is known about the biochemical properties of the isolated C2 domain. Since *MYBPC3*^Δ*25bp*^ alone is not a penetrant factor for the development of HCM and has been characterized before [45,46,47,48], our study focuses on the impact of the D389V mutation in the C2 domain of MyBPC. Residue D389 lies in a region of the C2 domain that is highly conserved across mammals (Figure 1B). The IgI-like domain consists of 10 β-strands that form two β-sheets (Figure 1C,D).

Here, we used a multifaceted approach to elucidate the role of the C2 domain and determine which of the versatile regulatory mechanisms of MyBPC are perturbed by mutation D389V. Correlation-guided enhanced molecular dynamics sampling (CORE-MD II) was applied to study the conformational space of the C2 domain [49,50,51] (see Section 4 for more details). The method relies in part on the partitioning of the entire pathway into short trajectories that we refer to as instances. The CORE-MD II sampling within each instance is accelerated by adaptive path-dependent metadynamics simulations. The second part of the enhanced sampling MD approach involves kinetic Monte Carlo (kMC) sampling between the different configurations that are accessed during each instance. The combination of the partition of the total simulation into short nonequilibrium simulations and the kMC sampling facilitates the sampling of rare events of protein dynamics on long timescales without definitions of a priori reaction pathways and additional parameters. Therefore, CORE-MD II provides an acceleration factor of at least 100, as validated by folding simulations of sample peptides [51]. Additionally, we used biochemical approaches to probe the structure, thermostability, and protein–protein interactions of the N-terminal domains of MyBPC.

## 2. Results

### 2.1. Enhanced CORE-MD II Simulations of the Wild Type and the D389V C2 Domain

We performed enhanced correlation-guided CORE-MD II simulations on the C2 wild type (wt) and the D389V domain for 100 ns each [49,50,51]. In Appendix A, we display relevant conformers that we extracted from the simulations at different points in time. In the visual analysis of the conformations, we observe that the N-terminal triple β-sheet structure remains relatively stable throughout the simulation in both C2 wt and D389V, whereas the C-terminal region of both proteins exhibits a stronger flexibility and populates a wider conformation range.

We continued with the analysis of the region in proximity to position 389. We determined the associated free energy landscapes as a function of the interatomic distances between D389 Cγ (wt)/V389 Cβ–K395 Nζ and D389 Cγ (wt)/V389 Cβ–A417 Cα averaged over the simulation. In Figure 2, we show the free energy landscapes determined for C2 wt and D389V. In the free energy landscape analysis of the wild type, we find four major minima surrounded by a region that is higher in free energy. States (1–3) reside at a distance D389–A417 of approximately 1.1 nm. State (1) corresponds to the formation of a salt bridge between D389 and K395 (see Figure 2A). States (2–3) represent intermediate states. State (4) is defined by a strong interaction of the D389 backbone with both the amino group of K395 and the backbone carbonyl group of the adjacent residue A417.

A sample conformer of State (4) is displayed in Figure 2D. The C2 D389V free energy landscape (Figure 2B) and the difference map between C2 wt and D389V (Figure 2C) reveal common features and differences in the population of conformational states. Predictably, State (1) and State (2) are not populated for C2 D389V, as their emergence relies on the stabilizing effect of the salt bridge formed between D389 and K395. State (3′) of C2 D389V resembles State (3) of wt but is characterized by the presence of additional hydrophobic contacts. It is the favorable conformation for C2 D389V, as it features a broad continuum of low energy states around its minimum. States (4) of wt and (4′) of D389V share similar coordinates in the free energy landscape, which shows that the mutation of the amino acid side chain does not significantly affect backbone interactions of residue 389.

Free energy landscape analysis reveals that mutation D389V leads to a shift in the conformational equilibrium and the annihilationof States (1–2). We investigated that effect through a kinetic network analysis (see Section 4). Figure 3 shows representative clustered conformations that were selected based on the distances of D389 Cγ or V389 Cβ to K395 Nζ and A417 Cα. The analysis of C2 wt shows that all states have a similar occupancy. State (2) mediates the transitions between States (1), (3), and (4), as indicated by their transition probabilities with State (2). The D389V mutation prevents the occupancy of States (1) and (2), leading to a high prevalence of State (3′) and a slight increase in the population of State (4′), compared to State (4) for C2 wt. Hence, the local perturbation by mutation D389V appears to have an allosteric effect that involves the occupancy of the energetic minima together with the associated kinetic network of state transitions, resulting in a reduction in the overall stability of the C2 domain [52,53].

To elucidate the effect of D389V on the global conformation of the C2 domain in more detail, we determined time-averaged distance maps for C2 wt and D389V (Figure 4A,B). Here, the main differences between C2 wt and D389V manifest themselves in relation to the β-strand contacts B4–B5–B6 (β_4–5–6_) and B1–B6 (β_1–6_) (see Figure 1D). C2 wt exhibits a tight network of interactions involving residues L421–E451 in β_4–5–6_, while the number of corresponding interactions is substantially reduced in C2 D389V (dotted boxes in Figure 4A,B). In particular, residues L421–D431 are no longer located near residues C436–E441, indicating destabilization or loss of critical β-strand contacts. The structural consequences of these observations are illustrated in Figure 4C,D for wt and D389V, respectively. A sample conformer of wt protein is shown in grey, and D389V is shown in blue, with both having residues D430–K450 colored in red. The C-terminal region of C2 wt is predicted to populate a larger ensemble of conformer structures compared to the initial NMR ensemble, where stabilizing interactions remain largely intact. For C2 D389V, residues D430–K450 adopt conformations that have a lower propensity for the formation of contacts β_4–5–6_. Furthermore, contacts between residues Q401–S406 and residues in the proximity of K380 are weakened, while new contacts are established between residues A392–L397 with the N-terminal region. Additionally, the missing salt-bridge interaction destabilizes the interaction between the N-terminal strand B1 and C-terminal strand B6, as indicated by the dashed boxes in Figure 4A,B. In summary, the effects implied by CORE-MD II simulations make it likely that protein thermodynamic properties are altered. We followed this lead by performing the pertinent biochemical experiments.

### 2.2. Structural Integrity and Thermodynamic Stability of N-Terminal MyBPC Domains

Proteins were prepared as described in Section 4, and purity was assessed by densitometric analysis of a Coomassie Brilliant Blue-stained SDS gel (Appendix A). β-Cardiac myosin (β-CM) and α-cardiac actin were approximately 95% and 98% pure. The MyBPC C2 and C0–C2 constructs of wt and D389V had a purity of >99%.

Depending on their character and position in the protein, point mutations may perturb protein structure by different mechanisms, such as impairment of folding pathways or loss of stabilizing side-chain interactions. We performed circular dichroism (CD) experiments to assess the impact of mutation D389V on C2 secondary structure composition. Moreover, we probed C2 domain thermostability using the thermal shift assay (TSA) and differential scanning calorimetry (DSC). The values for the secondary structure composition and melting temperatures of the MyBPC C2 and C0–C2 wt and D389V constructs are summarized in Table 1. CD spectra of the averaged data for each construct and the respective TSA and DSC traces are shown in Appendix A. CD, TSA, and DSC data indicate that all protein constructs possess a stable fold. CD spectra show that the C2 wt domain consists of 35–40% β-sheets and 30–35% unordered regions with approximately 10% α-helical and 20% β-turn content. These data are in good agreement with published NMR structures [31]. Our CD data indicate no major structural perturbations of the N-terminal domains of MyBPC by the D389V mutation.

Representative melting curves of thermal denaturation experiments are shown in Appendix A. The determined melting temperature (T_m_) values for the C2 wt domain (59.3 ± 0.1 °C in TSA and 56.5 ± 0.3 °C in DSC) are similar to those determined for the MyBPC C3 and C4 domains that exhibit the same IgI-like fold as the C2 domain [54,55]. The reduction in the T_m_ for the mutant C0–C2 construct shows that perturbations introduced in C2 can affect the thermal stability of neighboring N-terminal domains.

### 2.3. Interaction of N-Terminal Domains of MyBPC with Key Sarcomeric Proteins

In order to evaluate the binding affinity of MyBPC constructs C2 and C0–C2 to F-actin and synthetic thick filaments of β-CM, we performed high-speed cosedimentation assays. The effect of the interaction of MyBPC with the cardiac thin filament has been studied in detail, leading to the hypothesis that it modulates the actomyosin cross-bridge cycle [13,15,16,17]. In particular, the N-terminal MyBPC C0 domain was shown to bind the regulatory light chain of β-CM [56], and constructs comprising domains C0 to C2 were shown to interact with the S2 region of β-CM [18,19,57]. The degree to which C2 D389V alters the interactions between the N-terminus of MyBPC and cardiac thin filaments or β-CM may thus provide a direct indication of the extent of cardiac dysregulation.

High-speed cosedimentation assays probing the affinity of MyBPC C2 to actin or β-CM show very weak interaction (Appendix A). Cosedimentation of 30 µM C2 wt and 40 µM F-actin or 30 µM C2 wt and 2.5 µM β-CM resulted in <2% of C2 in the pellet fraction. Based on these results, a K_d_ in the range of several 100 µM can be estimated. This very weak binding is consistent with a K_d_ of 1.1 mM determined for the interaction of C2 with domain S2 of myosin [31].

To determine whether the D389V mutation affects the binding of other N-terminal domains to β-CM and actin, we performed cosedimentation assays with MyBPC C0–C2 (Figure 5). The K_d_ value determined for the interaction of C0–C2 with F-actin of 3.2 ± 0.9 µM is unchanged by the D389V mutation (2.8 ± 0.6 µM). The K_d_ value of the interaction of β-CM with C0–C2 wt has been calculated to be 12.7 ± 4.7 µM, whereas C0–C2 D389V has an affinity of 14.8 ± 7.7 µM. The values are consistent with prior studies [16,18,58,59] and indicate no effect of D389V on the interaction of the N-terminal region of MyBPC with F-actin and β-CM. Since the binding of C0–C2 to β-CM and F-actin is at least two orders of magnitude stronger, it is unlikely that the interactions of the C2 domain alone are of physiological importance. In fact, several models based on the results obtained with the C0–C2 construct suggest that the C2 domain does not make direct contact with F-actin [17]. Furthermore, it was shown that a C2–C5 construct does not incorporate into the sarcomere [18].

### 2.4. Simulation of C2 Domain Unfolding Dynamics

Apart from affecting the equilibrium between myosin disordered relaxed and super-relaxed states [19,60,61,62], MyBPC regulates contractility by acting as a mechanical tether. Electron tomography of cryopreserved muscle specimens shows that MyBPC is not only bound to the thick filament, but also extends out and connects to the thin filaments [63]. The C-terminal region of MyBPC is anchored to thick filaments via domains C8–C10. Additional binding of the N-terminal region of MyBPC to the thin filament establishes a mechanical tether that exerts a viscous load during the cross-bridge cycle [26,27,28,55]. Using laser-trap assays, the lifetimes of interaction between MyBPC and thin filaments were determined to range from 20 to 300 ms [28], outlasting actomyosin cross-bridge interactions (<200 µs) by at least two orders of magnitude [55,64]. Mutations that disrupt the mechanical tether function of MyBPC by reducing the folding stability of individual domains may thereby lead to altered sarcomere activity and thus contribute to HCM pathogenesis.

So far, it remains to be determined to what extent the N-terminal domains of MyBPC are contributing to the interaction with F-actin. While neutron scattering data suggest that domains C0 and C1 bind directly to F-actin [17], high-speed cosedimentation and yeast two-hybrid assays have shown that domains C1 and M are essential for the interaction [16,65]. As C8 is the closest domain attached to the thick filament, the intervening linker domains C2–C7 are predestined to unfold when force is applied during active sarcomere contraction. Using an atomic force microscopy-based approach, this model has been applied to HCM mutations R495W and R502Q in the C3 domain of MyBPC, which shares a similar IgI-like fold with domain C2. The unfolding force and the dynamics of unfolding and refolding were examined [55]. Since the C2 domain does not serve as an attachment site on thin filaments, it seems plausible that its function is to act as a brake exerting a set counterforce to cardiac contraction.

To model the mechanical strain to which the C2 domain is subjected, we applied steered molecular dynamics (SMD) simulations [66,67,68] by pulling on the Cα atom of P452 with constant force while S362Cα remained in place (Figure 6). Example traces of force–time curves are shown in Figure 6A. The maximum counterforce occurs at about 1100 ps for C2 D389V and at 2000 ps for C2 wt. This peak corresponds to the loss of the β_1–6_ interstrand contacts, as shown in exemplary structures at time points 0, 1400, and 2400 ps in Figure 6B. While the contact β_1–6_ is intact at t = 0 in both proteins, it is already broken at t = 1400 ps in C2 D389V. The disruption of this structure in C2 wt happens later, after its major peak, as visualized at t = 2400 ps. The destabilizing effect of the D389V mutation on the β_1–6_ contacts has already been established by the CORE-MD II simulation (Figure 4). Plots showing the residue matrices facilitate a comparison of the results obtained with both simulation methods and clarify their consistency (Appendix A and Figure 4).

The averaged force–time curves of 10 individual unfolding simulations are shown in Figure 6C. C2 D389V tends to unfold earlier with a sharper force peak and a higher maximum force. The same force is required to unfold C2 wt, but over a longer period of time. This suggests that this part of the C2 D389V structure is more rigid, whereas the C2 wt domain compensates for the applied force by forming various stabilizing contacts during the unfolding process. There were no noticeable differences in the force–time curve for the remaining protein unfolding pathway. Evaluating the time point of the first unfolding peak for each of the 10 experiments reveals that the C2 wt unfolding peak occurs after 1751 ± 714 ps and 2-fold faster for C2 D389V at 986 ± 384 ps (Figure 6D). Applying this unfolding mechanism to the MyBPC viscous load model on sarcomeric contraction implicates a different contraction-over-time pattern for carriers of the D389V mutation, as the C2 D389V domain exhibits different mechanical spring properties than C2 wt. Disruption of the finely tuned cardiac contractile cycle provides a molecular mechanism for the development of HCM. These results provide the basis for further clarification of the pathomechanism of double mutation *MYBPC3*^Δ*25bp/D389V*^.

## 3. Discussion

Mutations in the *MYBPC3* gene are among the most common causes for the development of HCM. In this study, we demonstrate that the D389V mutation strongly perturbs the conformational space of the C2 domain, while leaving the tertiary structure intact. Residue D389 of MyBPC is conserved across many species and isoforms, which appears to be related to its role in the formation of a salt bridge that stabilizes the loop structure in which it resides (Figure 1). Four energetic minima were identified by the use of CORE-MD II simulations, involving the interaction of D389 with K395 and A417 (Figure 2 and Figure 3). Mutation D389V abolishes the electrostatic attraction towards K395, required for the formation and stabilization of States (1) and (2). Consequently, the energy landscape of C2 is distorted and State (3′) becomes the predominant state in C2 D389V. This allosteric trigger event propagates throughout the whole domain and destabilizes the β_4–5–6_ and β_1–6_ interstrand contacts (Figure 4).

The replacement of single amino acids in MyBPC can in some instances disrupt the tertiary structure of the affected domain, as shown for the A31P mutation in the C0 domain [41]. Our CD data suggest that the D389V mutation does not cause a major structural change in the domain structure of the constructs C2 and C0–C2 (Table 1 and Appendix A). However, the thermostability of the C2 and C0–C2 domain constructs is decreased by D389V, as demonstrated by TSA and DSC (Table 1), which indicates a different conformational space occupied by these constructs compared to wt constructs. This is in line with the results that were obtained by CORE-MD II simulations.

To fulfill its regulatory roles, MyBPC depends on the interaction of its N-terminal domains with the key sarcomeric proteins actin and β-CM [13,15,16,17,18,19,57]. Our high-speed cosedimentation data on the isolated C2 domain in interaction with β-CM and actin suggest very weak binding (Appendix A), which is consistent with a K_d_ value of 1.1 mM previously determined for the interaction of C2 with the region containing R870 within subfragment-2 of β-CM [31]. The affinities determined for the interaction of C0–C2 with β-CM and actin are at least two orders of magnitude greater, which implies that potential interactions of C2 make a minor contribution compared to those of the other N-terminal domains. The values are unchanged by the D389V mutation and consistent with those determined in previous studies [16,18,58,59], thus excluding altered affinities as a pathomechanism.

More recently, the paradigm of MyBPC acting as a mechanical tether has come into focus [69]. Studies exposing MyBPC to mechanical strain by the use of atomic force microscopy revealed a structural hierarchy of domain unfolding that is highly conserved across species [26,27]. Our data suggest that C2 is a linker domain that contributes a resisting force during cardiac contraction. A similar mechanism has been proposed for the neighboring C3 domain that harbors several HCM mutations [55]. SMD simulations were used to explore potential differences in the unfolding pathways of C2 wt and D389V by moving the C-terminus and fixing the N-terminus (Figure 6 and Appendix A). Compared to C2 wt, β_1_–_6_ contacts are disrupted twice as fast and with a sharper peak in C2 D389V. In addition, using an enhanced MD simulation method, we found that the weakening of β_1_–_6_ interactions is a direct consequence of mutation D389V (Figure 4).

There are several possibilities for how these data relate to the hyperdynamic findings observed in echocardiograms of *MYBPC3*^Δ*25bp/D389V*^ carriers [48]. The mechanical destabilization of the C2 domain by the D389V mutation results in a reduced capability of MyBPC to apply a resisting force. Moreover, a differential expression of wt and mutant allele across cardiomyocytes, as demonstrated for HCM mutations in *MYH7* [70,71,72], can promote an imbalance in contractility of the myocardium.

MyBPC can only exert its tethering function if it is correctly incorporated into the C-zone of the sarcomere. Overproduction of MyBPC^C10mut^ in rodents has been shown to lead to mislocalization of the protein in Z-discs, contractile dysfunction, and an HCM-like phenotype [47]. The pathogenic effects associated with *MYBPC3*^Δ*25bp*^ increase with age. Young and middle-aged individuals carrying the deletion are mostly asymptomatic or show only mild hypertrophy, while 90% of the *MYBPC3*^Δ*25bp*^ carriers that are older than 60 become symptomatic [45]. In contrast, *MYBPC3*^Δ*25bp/D389V*^ carriers exhibit hyperdynamic systolic function with high penetrance already at a young age. These comprise an increased left ventricular ejection fraction, abnormal calcium transients, and cardiomyocyte hypertrophy, features not inherent to *MYBPC3*^Δ*25bp*^ carriers [48]. Unless the mutation *MYBPC3*^Δ*25bp/D389V*^ leads to the incorporation of mutant protein into the contractile zone of sarcomeres, there is no plausible molecular mechanism to explain the differences between *MYBPC3*^Δ*25bp*^ and *MYBPC3*^Δ*25bp/D389V*^ carriers. Many of the HCM mutations that are associated with *MYBPC3* are frameshift mutations and result in the formation of truncated proteins. The 25 base pair deletion in intron 32 of *MYBPC3*^Δ*25bp*^ and *MYBPC3*^Δ*25bp/D389V*^ has been shown to produce mRNA splice variants that lack exon 33. The probability of this event has been suggested to depend on additional risk factors, such as metabolic syndrome or environmental factors [73]. Moreover, alternative splicing has been shown to be a stochastic process, resulting in a range of differently processed mRNA transcripts [74,75]. In fact, this has already been demonstrated for the cardiomyopathy-related gene *LMNA* [76] and *MYBPC3* [77]. New splice isoforms were found in patients with the *MYBPC3* c.1898-1G>A splice site alteration, although the canonical mRNA sequence was still the predominant one. Truncated proteins produced as a result of frameshift mutations can potentially exert a poison polypeptide effect. However, because they cannot be productively integrated into sarcomeric structures, they are usually efficiently degraded by the ubiquitin–proteasome system in cardiomyocytes, resulting in a haploinsufficiency phenotype [32,33,34,35,36,37]. In contrast, single amino acid exchanges in MyBPC reduce the folding stability of affected structural domains without impeding their integration into the contractile zone of sarcomeres. Their increased propensity to irreversibly unfold under the mechanical stress of the contracting sarcomere suggests a poison polypeptide effect as the dominant pathomechanism for MYBPC^D389V^ in young and middle-aged mutation carriers [55,78,79,80].

Differences in the progression of HCM between *MYBPC3*^Δ*25bp*^ and *MYBPC3*^Δ*25bp/D389V*^ carriers and age-dependent changes in splicing patterns [81,82,83,84,85] suggest a shift in the dominant pathomechanism for elderly *MYBPC3*^Δ*25bp/D389V*^ carriers. The consequences of mutations *MYBPC3*^Δ*25bp*^ and *MYBPC3*^Δ*25bp/D389V*^ for individuals with a single mutant allele with respect to disease mechanism and the clinical phenotype are schematically depicted in Figure 7. Young carriers exhibit no symptoms or mild conditions because they predominantly produce the wt protein originating from the correctly spliced mRNA. With increasing age, the amount of mutant transcript with skipped exon 33 increases, leading to the production of MyBPC^C10mut^, which triggers HCM by C-zone haploinsufficiency [47]. Roughly 90% of *MYBPC3*^Δ*25bp*^ carriers above the age of 60 are symptomatic for HCM [45].

Young and middle-aged *MYBPC3*^Δ*25bp/D389V*^ carriers produce the protein with a D389V mutation in the C2 domain and an intact C10 domain. The protein localizes correctly in the C-zone, where its function is impaired due to its reduced folding stability and sensitivity to mechanical stress. The resulting toxic buildup of misfolded protein impairs sarcomere function. Such a mechanism is compatible with clinical hyperdynamic findings associated with HCM that are observed in *MYBPC3*^Δ*25bp/D389V*^ carriers [48]. With increasing age, as for the *MYBPC3*^Δ*25bp*^ carriers, changes in alternative splicing result in a gradual shift in the production of splice isoforms that encode truncated gene products, which are in part degraded by the ubiquitin–proteasome system or mislocalize to the Z-disc. The main contributing factors to the pathomechanism in older *MYBPC3*^Δ*25bp*^ and *MYBPC3*^Δ*25bp/D389V*^ carriers appear therefore to be identical, namely impaired sarcomere function due to C-zone haploinsufficiency with contributions from the mislocalization of truncated protein to the Z-disc.

The genotype–phenotype correlations of MyBPC are very diverse due to the multiplicity of genes and environmental factors influencing the clinical outcome. For example, the occurrence of an *MYH7* mutation in *MYBPC3*^Δ*25bp*^ carriers frequently leads to sudden cardiac death [43]. Extensive genetic screening approaches can discover additional risk factors for genetic variants with an incomplete penetrance for the development of HCM, as has been demonstrated for the *MYBPC3*^Δ*25bp/D389V*^ variant [48]. The pathomechanism largely depends on the ratio of wt to alternatively spliced mRNA transcript originating from the *MYBPC3*^Δ*25bp*^ allele. This question can be addressed by real-time quantitative PCR (qPCR) on tissue biopsies of *MYBPC3*^Δ*25bp*^ and *MYBPC3*^Δ*25bp/D389V*^ carriers using mRNA transcript-specific primers. On the protein level, mass spectrometry can provide information on whether and how much MyBPC^C10mut^ or MyBPC^D389V/C10mut^ protein is produced in the patients. Understanding the molecular details in the pathogenesis of this life-threatening disease will improve clinical outcome predictions and aid in the development of appropriate treatment regimens.

## 4. Materials and Methods

### 4.1. Homology Modeling of the C2 Domain

Homology models of human MyBPC C2 domain were created from an NMR structure of the M and C2 domain (PDB accession code 5k6p [86]) using the software MODELLER [87]. Residues 362–452 of human MyBPC (UniProt Q14896) were aligned to the template using the align2d command. The resulting alignment file served as an input for the AutoModel class, which calculates five 3D models. The model with the highest score was chosen for simulations. The D389V mutation was inserted using the mutator plugin of PyMOL 1.8.6.2 (Schrödinger Inc., New York, NY, USA).

### 4.2. CORE-MD II Simulation

The CORE-MD II technique is an enhanced MD-simulation method that relies in part on the partitioning of the entire pathway into short trajectories that we refer to as instances. The CORE-MD II sampling within each instance is accelerated by adaptive path-dependent metadynamics simulations. The second part of the enhanced sampling MD approach involves kinetic Monte Carlo (kMC) sampling between the different configurations that are accessed during each instance. The combination of the partition of the total simulation into short nonequilibrium simulations and the kMC sampling facilitates the sampling of rare events of protein dynamics on long timescales without definitions of a priori reaction pathways and additional parameters.

For the derivation of the CORE-MD II method, a global probability $P\left(x_{i}\left(t\right)\right)$ is defined that can be subdivided over *N* time-slices or subtrajectories *k* with length $\tau_{k}$, which are described by local probability densities $\rho_{k}\left(x_{i}\left(t\right)\right)$:(1)$$\left(x_{i}\left(t\right)\right)=\underset{N\to \infty }{lim}\overset{N}{\underset{k}{\prod }}\rho_{k}\left(x_{i}\left(t\right)\right)$$
where $x_{i}\left(t\right)$ stands for the coordinate of an atom with the index i.

In the CORE-MD II formalism, we then consider the averaging process of a trajectory-dependent quantity X(t); the partition into small trajectories allows for a faster formation of time averages than the determination of the expectation value of the complete trajectory, which is linked to the timescale problem of MD simulations. Therefore, the expectation value of the complete trajectory can be approximated as:(2)$$\langle X\left(t\right)\rangle =X\left(t\right)P\left(x_{i}\left(t\right)\right)\approx X\left(t\right)\overset{K}{\underset{k}{\prod }}\rho_{k}\left(x_{i}\left(t\right)\right)$$
which states that the partition of the complete trajectory into a finite number of K subtrajectories is approximately sufficient for the sampling of the expectation value ⟨X(t)⟩. We define the number of configurations K by a minimal set of the number of atoms *N_a_* in the system, which guarantees a fast forward propagation.

Within each instance k, the local pathway is described by the reduced action $L_{i_{k}}\left(t\right)$:(3)$$L_{i_{k}}\left(t\right)=\underset{t<\tau_{k}}{\sum }p_{i}\left(t\right)\Delta x_{i}\left(t\right)$$
where $\Delta x_{i}\left(t\right)=x_{i}\left(t\right)-\langle x_{i}\left(t\right)\rangle$, $p_{i}\left(t\right)$ is the momentum, and t stands for the time. The local path $L_{i_{k}}\left(t\right)$ is used to define the local autocorrelation function $C_{i_{k}}\left(t\right)$:(4)$$C_{i_{k}}\left(t\right)=\frac{1}{\tau_{k}}\underset{t\leq \tau_{k}}{\sum }\frac{\left(L_{i_{k}}\left(t\right)-\langle L_{i_{k}}\left(t\right)\rangle \right)\left(L_{i_{k}}\left(t^{\prime }\right)-\langle L_{i_{k}}\left(t\right)\rangle \right)}{\left|L_{i_{k}}\left(t\right)-\langle L_{i_{k}}\left(t\right)\rangle \right|\left|L_{i_{k}}\left(t^{\prime }\right)-\langle L_{i_{k}}\left(t\right)\rangle \right|}$$
where $L_{i_{k}}\left(t^{\prime }\right)$ is determined with a frequency equal to 1 ps^−1^ and ⟨…⟩ denotes the time average.

The CORE-MD II [51] technique samples the system along a correlation-dependent probability between states with an index *k* using a kinetic Monte Carlo (kMC) algorithm. We limit the number of kMC configurations by a minimal set of the number of atoms $N_{a}$ in the system, which guarantees a fast forward propagation within a small window of three possible selections in each kMC step.

With a frequency of $\tau_{k}$*,* we perform a kMC step and express a rate $r_{k}$ for each instance *k*. We then calculate the cumulative rates and apply the kinetic Monte Carlo algorithm for the selection of a configuration k with which a configuration is used for the subsequent trajectory instance. The kMC sampling guides the trajectory between equilibrium configurations of the system, where each instance k resembles a state that resides close to the equilibrium state.

We continue with the description of the second component of the CORE-MD II algorithm that applies the local biases. (1) At each initialization of a new trajectory fragment, the velocities are selected from a random distribution. (2) In order to accelerate the sampling within each instance, we apply a history-dependent bias potential that is related to metadynamics [88], while the history dependency is limited by the timescale of each instance. We add the Gaussians to the history-dependent potential using the well-tempered metadynamics technique through a normalization of the added Gaussians by the factor. The corresponding bias is added throughout the simulation.

Finally, we accelerate the sampling within each instance and apply the statistical bias as described in our recent work on the CORE-MD algorithm. We implemented the correlation-dependent bias by a factorization with the variable $\lambda_{i_{k}}\left(t\right)$ with which we scale the gradient of all atoms in the system. This statistical bias enhances the decay of the correlation function and accelerates the access of new states by the system.

For the enhanced MD simulations and the trajectory analysis, we used a modified version of the GROMACS 4.5.5 simulation package [49,50,51]. For the setup of our simulations, we centered the homology model of the NMR structure (PDB: 5k6p) in a cubic box with dimensions 5.75 × 5.75 × 5.75 nm^3^. For the description of the interactions in the system, we used the AMBER99SB force field [89,90]. The simulations have been performed in implicit solvent (GBSA), while the electrostatic and van der Waals (vdW) interactions were calculated using a twin-range cutoff of 1.2/1.0 nm. The neighbor list with a neighbor list cutoff equal to 1.0 nm was updated every second integration step. The simulations were carried out in the constant-NVT ensemble at a target temperature equal to 300 K. For temperature control, we used the Nosé–Hoover thermostat with a coupling constant τ_T_ equal to 1.0 ps.

### 4.3. MD Trajectory Analysis

In order to define the free energy landscapes, we projected the free energy on the interatomic distances that we identified as principal coordinates (distance definitions). We defined the free energies ΔF using the relation [91]:(5)$$\Delta F=k_{b}T\frac{P_{i}}{P_{min}}$$

To compare the free energy landscapes of two conformational distributions, we calculated the free energy differences ΔΔF:(6)$$\Delta \Delta F=\Delta F_{1}-\Delta F_{2}$$

Using the same distance definition, we performed a clustering of states with a cutoff of 0.2 nm, in order to determine whether a conformation belongs to a state. We calculated the transition probabilities and the probability for the occurrence of a state using the relative counts of each cluster in relation to the total number of conformations within each state.

### 4.4. Plasmid Construction

C0–C2 and C2 constructs of MyBPC (UniProt Q14896) with a C-terminal 6xHis-tag were generated from the human full-length cDNA. Gene fragments derived from conventional PCR were inserted between the NdeI and EcoRI sites of pET23a(+). C0–C2 comprises the residues 1–452 and C2 the residues 362–452, consistent with the UniProt entry. D389V variants were generated using site-directed whole plasmid mutagenesis.

### 4.5. Protein Purification

Human MyBPC (Uniprot Q14896) constructs C2 and C0–C2 were produced in *Escherichia coli* Rosetta(DE3)pLysS cells. Purification was performed as described in previous studies [15,16]. MyBPC C2 and C0–C2 were affinity-purified using PureCube 100 Ni–NTA Agarose (Cube Biotech, Monheim am Rhein, Germany). Protein purity was assessed using SDS-PAGE [92], and protein concentration was determined photometrically using calculated extinction coefficients at 280 nm of 10,095 L mol^−1^ cm^−1^ for C2 and 42,650 L mol^−1^ cm^−1^ for C0–C2 [93,94]. C2 was subsequently applied to a Superdex 16/600 S75 size exclusion chromatography column (GE Healthcare, Chicago, IL, USA). C0–C2 was further purified using a HiTrap SP HP 5 mL cation exchange column (Cytiva Life Sciences, Marlborough, MA, USA). Trehalose was supplemented to 10% (*w/v*); protein aliquots were flash-frozen in liquid N_2_ and stored at −80 °C until usage.

β-CM from *Sus scrofa* left ventricular tissue (Uniprot P79293, Q5EFJ2, Q8MHY0) was purified as described in [95]. α-Cardiac actin (Uniprot B6VNT8) was extracted from acetone powder of *Sus scrofa* cardiac tissue according to [96].

### 4.6. Thermal Stability of MyBPC Subdomains

Protein thermostability was explored in a thermal shift assay (TSA) [97] using SYPRO Orange Protein Gel Stain (Thermo Fisher Scientific, Waltham, MA, USA). C2 was diluted to a concentration of 0.2 mg/mL in a buffer containing 10 mM HEPES pH 7.4, 100 mM NaCl, 1 mM EDTA, and 1 mM DTT. C0–C2 was adjusted to 0.5 mg/mL in the same buffer containing 150 mM NaCl. Protein solution was subjected to a temperature gradient of 25 to 80 °C and fluorescence was monitored using a StepOne Real-Time PCR System (Applied Biosystems, Waltham, MA, USA). Protein melting temperature (T_m_) was determined as the midpoint of fluorescence increase.

Differential scanning calorimetry (DSC) was performed using a Nano DSC Model 6300 Differential Scanning Calorimeter (TA Instruments, New Castle, DE, USA). C0–C2 was diluted to 1.5 mg/mL and C2 to 3.0 mg/mL in the buffers mentioned above. The protein solutions were subjected to a temperature gradient of 25 to 70 °C and the amount of heat required to increase temperature was monitored. The maximum of the acquired curve corresponded to T_m_. Data were analyzed using NanoAnalyze software (TA Instruments, New Castle, DE, USA).

### 4.7. Circular Dichroism Measurements

Circular dichroism measurements were performed using PiStar 180 CD spectrometer (Applied Photophysics, Leatherhead, UK). Proteins were diluted into 10 mM NaP_i_ buffer pH 7.4 with 0.2 mg/mL MyBPC C0–C2 and 0.1 mg/mL MyBPC final concentrations. Circular dichroism was recorded as millidegrees for wavelengths from 260 to 176 nm with a bandwidth of 4 nm for buffer and protein solutions. Data for each wavelength were obtained for 10 s and each protein was measured as a triplicate. Data were averaged and buffer subtracted before smoothening using the Savitzky–Golay filter with a polynomial power of 2. Preprocessed data were analyzed using the DichroWeb server [98] and a set of reference spectra for wavelengths from 185 to 240 nm [99]. The output of the analysis was the mean residue ellipticity for each wavelength and the resulting calculated relative amounts of α-helical, β-sheet, and unordered regions.

### 4.8. High-Speed Cosedimentation Assay

High-speed cosedimentation assays were used to study protein–protein interactions as described previously [100,101]. G-actin was polymerized for 3 h at room temperature by the addition of MgCl_2_ to 2 mM and KCl to 100 mM. F-actin (0–40 µM) was incubated with 5 µM C0–C2 or 30 µM C2 in 25 mM HEPES pH 7.3, 25 mM KCl, 5 mM MgCl_2_, and 1 mM EGTA for 30 min at room temperature. Reaction setups were centrifuged for 30 min at 55,000 rpm using a TLA-55 rotor in an Optima TLX Ultracentrifuge (Beckman Coulter, Brea, CA, USA). Protein content in supernatant and pellet samples was analyzed using SDS-PAGE and densitometric quantification with Image Lab (Bio-Rad Laboratories, Hercules, CA, USA).

Synthetic thick filaments of full-length β-CM were created by adjusting buffer conditions to 20 mM HEPES pH 7.4, 75 mM NaCl, 2 mM MgCl_2_, 0.5 mM EDTA, and 0.5 mM DTT and incubation on ice for 30 min prior to the experiment. β-CM (2.5 µM) was incubated with 0–30 µM C0–C2 or C2 for 30 min at room temperature. Centrifugation was performed as above and pellet fractions were analyzed using SDS-PAGE. Amount of protein in the pellet fractions was determined via a protein standard range.

Data analysis was performed using Origin 2020b (OriginLab, Northampton, MA, USA).

### 4.9. Steered Molecular Dynamics Simulation

The mechanostability of C2 was assessed using constant velocity steered molecular dynamics (cvSMD) simulations [66,67,68] employing NAMD 2.14 [102] with CHARMM36 force fields [103]. The C2 domain was modeled as described above. Temperature was set to 310 K and the GBIS model was used with an alpha-cutoff of 1.2 nm and an ion concentration of 150 mM NaCl. Scaled1–4 was used for nonbonded interactions with a cutoff of 1.4 nm and a switching function at 1.0 nm. A Langevin Nosé–Hoover thermostat was applied with a damping constant of 5.0/ps and hydrogen coupling enabled. Atom restraints were applied to both the fixed and the SMD atom with an exponent for position restraint energy function of 2 and a constraint scaling factor of 1.0. Molecules were equilibrated for 1 ns before performing a 20 ns unfolding simulation. The SMD atom was moved at a constant velocity of 0.01 Å/ps with a force constant of 7 kcal/mol/Å equaling 486.4 pN Å over the course of 20 ns along the vector defined by the way from the fixed to the SMD atom. Results were analyzed by multiplying the force in x, y, and z directions acting on the SMD atom with the normalized direction of pulling and plotting the result against the time. Data were smoothened using Savitzky–Golay filter to compensate for the noise created by the spring acting on the fixed atom. Averaged force–time curves consist of 10 separate unfolding simulations per construct.

## Figures and Tables

**Figure 1 ijms-22-11949-f001:**
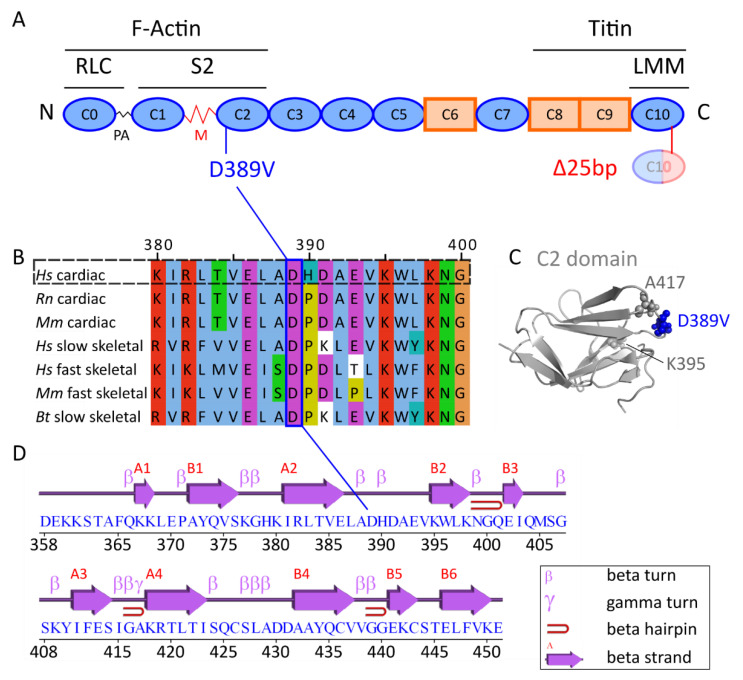
Schematic diagram showing the domain organization of MyBPC and its C2 domain. (**A**) MyBPC consists of eight IgI-like domains (blue), three FnIII-like domains (orange), and one unique M domain. Binding partners are indicated above the respective subdomains. The positions of the *MYBPC3*^Δ*25bp/D389V*^ mutation are indicated. PA: proline–alanine-rich region. (**B**) Alignment of MyBPC C2 protein sequences. Residue D389 is highly conserved across different mammalian isoforms. Rn: *Rattus norvegicus*, Mm: *Mus musculus*, Hs: *Homo sapiens*, Bt: *Bos taurus*. Alignment was created using T-Coffee [29] and Jalview [30]. (**C**) Cartoon representation of the MyBPC C2 domain (PDB: 5k6p [31]) with relevant residues D389, K395, and A417 displayed as spheres. (**D**) Domain topology scheme of the C2 domain as generated by EMBL-EBI PDBsum (https://bio.tools/pdbsum_generate; last accessed on 1 November 2021). A1–A4 and B1–B6 represent strands of the β-sheets A and B.

**Figure 2 ijms-22-11949-f002:**
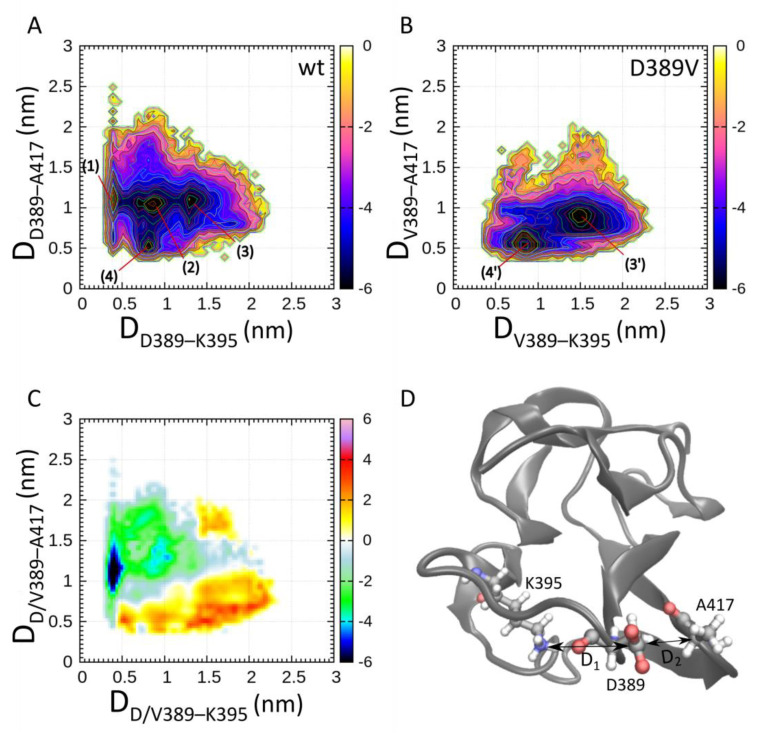
Free energy landscapes of MyBPC C2 wt (**A**) and D389V (**B**) domains generated using 100 ns of CORE-MD II simulations. The effect of the D389V mutation on the conformational space in the vicinity of the mutation is explored. We define the interatomic distances between D389V Cγ/V389 Cβ–K395 Nζ and D389V Cγ/V389 Cβ–A417 Cα as principal coordinates of the FEL. The scale of the color bar is given in units of k_B_T and represents the free energy ΔF (see Equation (5), Section 4.3). (1) to (4) indicate conformational states of C2 wt, and (3′) and (4′) indicate those of C2 D389V. (**C**) Free energy differences ΔΔF between C2 wt and C2 D389V (see Equation (6), Section 4.3). Negative values indicate an abundance of wt, and positive values indicate an abundance of D389V states. (**D**) Sample conformer extracted from State (4) in (**A**). C2 wt is depicted with the interacting residues D389, K395, and A417. D_1_ represents the x-axis and D_2_ represents the y-axis of the distances plotted in figures (**A**) to (**C**) for all conformations adapted over the course of the simulation.

**Figure 3 ijms-22-11949-f003:**
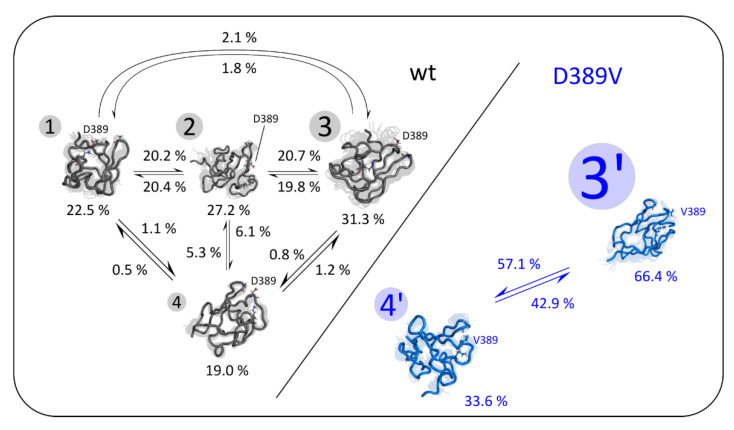
Kinetic network representation derived from the clustered major conformations of C2 wt and D389V. Representative conformations of the States (1) to (4) identified in Figure 2 are shown for C2 wt (black), and (3′) and (4′) are shown for D389V (blue). Backbone Cα is shown as a tube representation, and the side chains of residues D389, V389, K395, and A417 are shown as ball–and–stick models with CPK coloring. The opaque underlay displays 40 conformers of each state. The numbers with circular background denote different states. Their size corresponds to the relative occupancy of each state over the course of the simulation, which is shown below the respective conformers. Arrows and percentages reflect probabilities of transition between two states.

**Figure 4 ijms-22-11949-f004:**
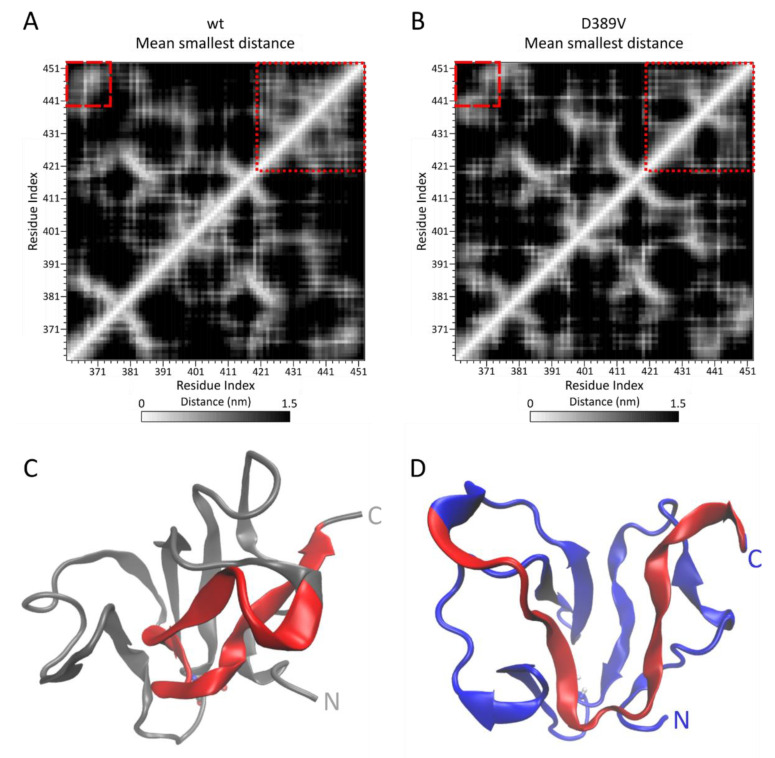
Mean smallest distance matrices for MyBPC C2 wt (**A**) and D389V (**B**). The time-averaged minimum distance of each residue to every other residue is plotted in heat maps with white indicating the smallest distance and black indicating the largest distance ranging from 0 to 1.5 nm. This map reveals differences in the global conformational space of C2 wt and D389V. Dotted boxes indicate the β_4–5–6_ interstrand contacts within the C-terminus of the domain. Dashed boxes indicate the β_1–6_ interstrand contacts of C2 wt that are destabilized in C2 D389V (for topology see Figure 1D). (**C**,**D**) Sample structures of C2 wt (grey) and D389V (blue) to illustrate the differences in C-terminal conformation regarding the dotted boxes in (**A**,**B**). Residues 430 to 450 are marked in red.

**Figure 5 ijms-22-11949-f005:**
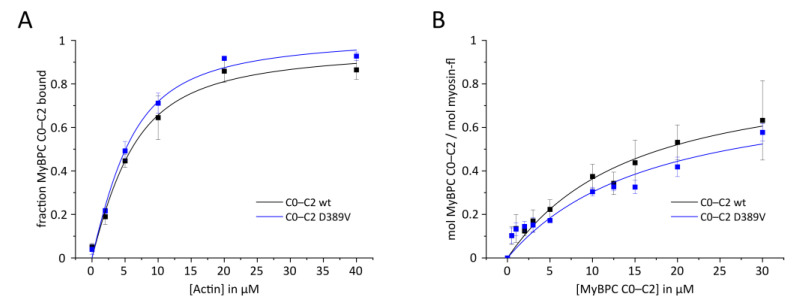
High-speed cosedimentation experiments of MyBPC C0–C2 with F-actin and β-cardiac myosin. (**A**) Actin (0–40 µM) and MyBPC C0–C2 (5 µM) were cosedimented and the fraction of C0–C2 in the pellet was determined. A quadratic fit yields the dissociation constants K_d_ for wt of 3.2 ± 0.9 µM and D389V of 2.8 ± 0.6 µM. (**B**) MyBPC C0–C2 (0–30 µM) was cosedimented with 2.5 µM of β-cardiac myosin. The amount of C0–C2 bound to synthetic thick filaments was determined using a protein standard range of C0–C2. A quadratic fit yields K_d_ values for wt of 12.7 ± 4.7 µM and D389V of 14.8 ± 7.7 µM. Data points indicate mean ± SD from at least *n* = 3 experiments.

**Figure 6 ijms-22-11949-f006:**
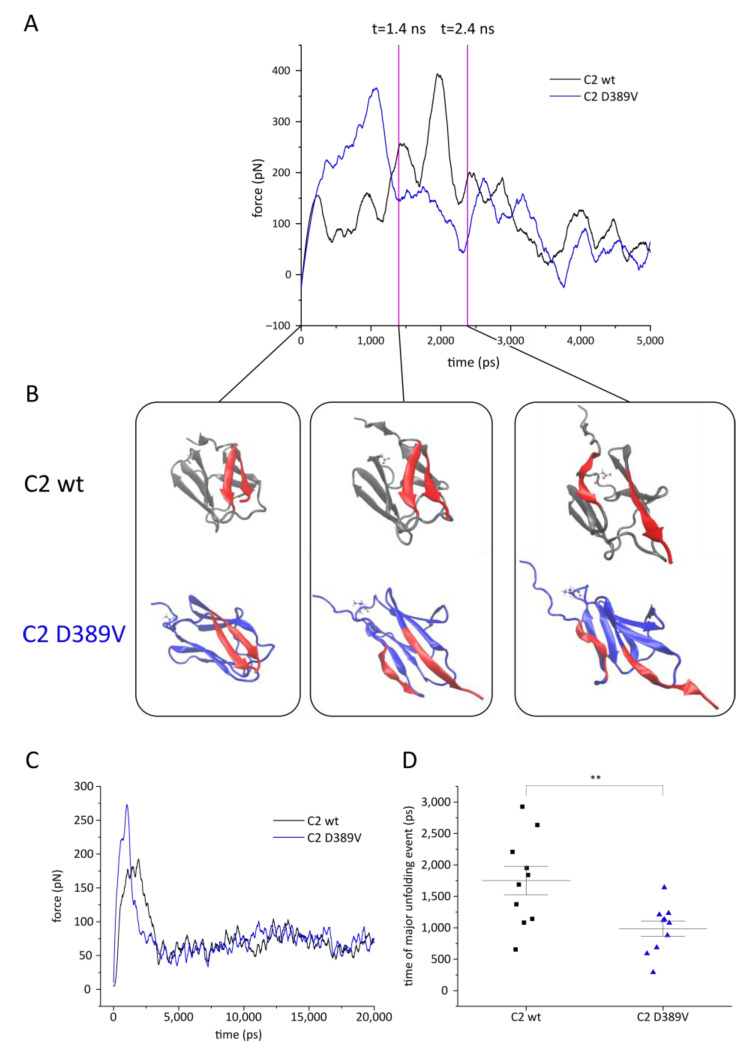
Constant velocity steered molecular dynamics (cvSMD) simulation of MyBPC C2 domain unfolding. The SMD atom P452Cα was pulled at a velocity of 0.01 Å/ps and the spring constant was set to 7 kcal/mol/Å. (**A**) Sample traces for an unfolding simulation of each wt and D389V MyBPC C2 domain. The force acting on the SMD atom is plotted against the time of the simulation. Data were smoothened using Savitzky–Golay filter to compensate for the noise created by the spring acting on the fixed atom. (**B**) Sample structures of time points 0, 1400, and 2400 ps in the unfolding experiments displayed in (**A**). C2 wt and C2 D389V are shown in grey and blue, respectively, with residue 389 as a CPK model. Residues A372–K377 and T445–P452 of the parallel β_1–6_ strands are marked in red. (**C**) Average of *n* = 10 independent unfolding simulations of each wt and D389V MyBPC C2 domain. The force acting on the SMD atom is plotted against the time of the simulation. Data were smoothened using Savitzky–Golay filter to compensate for the noise created by the spring acting on the fixed atom. (**D**) The time point of the major unfolding event is plotted for each individual unfolding experiment. The major unfolding event is defined as the global maximum of the force–time curve. The peak of unfolding occurred on average after 1751 ± 714 ps for wt and 986 ± 384 ps for D389V. ** *p* < 0.01.

**Figure 7 ijms-22-11949-f007:**
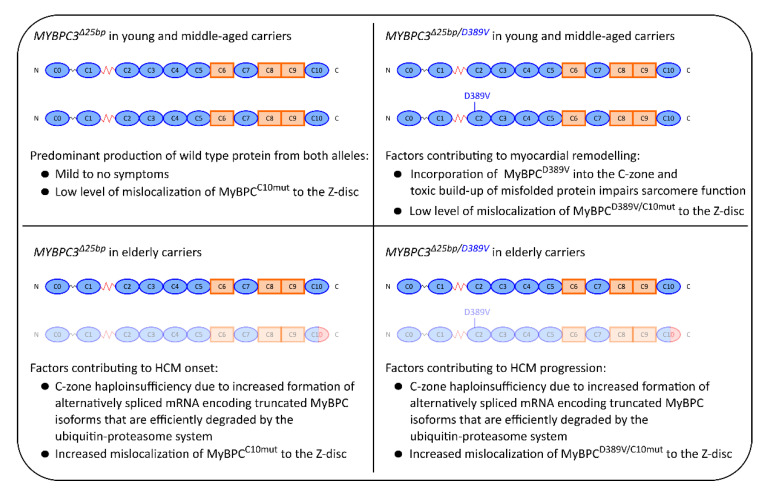
Proposed disease mechanisms as a result of age-related alternative splicing. All carriers in this model are heterozygous for one of the variants and produce wt protein from at least one allele at all times. Young and middle-aged *MYBPC3*^Δ*25bp*^ carriers predominantly produce wt mRNA from both alleles, resulting in mild or absent symptoms. Incorporation of MyBPC^D389V^ into the contractile zone of sarcomeres in young and middle-aged *MYBPC3*^Δ*25bp/D389V*^ carriers leads to myocardial remodeling mediated by a poison polypeptide mechanism. As patients age, the alternatively spliced transcript on the allele carrying the intron deletion predominates in both *MYBPC3*^Δ*25bp*^ and *MYBPC3*^Δ*25bp/D389V*^ carriers. We assume that the truncated forms of the protein are efficiently eliminated by the ubiquitin–proteasome system in cardiomyocytes with smaller fractions of MyBPC^C10mut^ and MyBPC^D389V/C10mut^ mislocalizing to the Z-disc. As a result, C-zone haploinsufficiency is predicted to become the dominant disease mechanism in elderly *MYBPC3*^Δ*25bp*^ and *MYBPC3*^Δ*25bp/D389V*^ carriers.

**Table 1 ijms-22-11949-t001:** Structural and thermodynamic analysis of N-terminal MyBPC domains as determined by CD experiments, TSA and DSC. α: α-helix, β: β-sheet, T: turn, U: unordered, Σ: total, T_m_: melting temperature. Data are mean ± SEM of at least *n* = 3 experiments.

Construct	α (%)	β (%)	T (%)	U (%)	Σ (%)	T_m_ (TSA)/°C	T_m_ (DSC)/°C
C2 wt	7	37	22	34	100	59.3 ± 0.1	56.5 ± 0.3
C2 D389V	7	36	22	35	100	52.1 ± 0.3	52.3 ± 0.2
C0–C2 wt	7	36	24	32	99	51.8 ± 0.5	50.2 ± 0.1
C0–C2 D389V	8	40	22	30	100	48.9 ± 0.6	47.6 ± 0.3

## Data Availability

Not applicable.

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
