# Peer review of "Assessment of the Contribution of a Thermodynamic and Mechanical Destabilization of Myosin-Binding Protein C Domain C2 to the Pathomechanism of Hypertrophic Cardiomyopathy-Causing Double Mutation MYBPC3Δ25bp/D389V"

_ijms, 2021, doi:10.3390/ijms222111949_

Round 1

Reviewer 1 Report

Hypertrophic cardiomyopathy (HCM) is the most frequent genetic cardiac disorders and MYBPC3 and MYH7 are the most prevalent genes involved. MYBPC3Δ25bp carrier is a common variant and contributes to HCM through haploinsufficiency theory. A new variant with monoallelic double mutation MYBPC3Δ25bp/D389V is a subset of the common MYBPC3Δ25bp variant. The deletion of 25 bp variant affects MYBPC domains nearing C8-C10, while the D389V localizes at the C2 domain. In this manuscript, the C2 domain of the MYBPC with wild type or D389V were evaluated and compared through modeling systems. The findings showed that D389V had little effect on its interaction with myosin/actin protein, instead, caused conformational configuration shift and altered thermostability.

Data of modeling evaluation provided hypothesis for the explanation of pathomechanism for the development of HCM in MYBPC3Δ25bp/D389V variant. Minor suggestion was as follows.

  1. In the current version, the major and final conclusion was not found in the abstract. A new version is necessary.
  2. MYBPC3Δ25bp is a common genetic variant for the development of HCM. MYBPPC3 D389V is a new variant found only in the subset of MYBPC3Δ25bp The D389V variant was not found alone. C2 domain with wild type or D389V was target for evaluation and comparison. It is curious to ask what findings in the presence of double mutations.

Author Response

We thank the reviewers and the editor for the constructive comments and valuable input to improve the quality of our manuscript.

We have addressed all concerns raised by the referees. All major modifications made to the manuscript are highlighted using the “track changes” feature. We also provide a clean copy to the submission.

Our responses to the reviewers' comments are shown in italics.

Reviewer #1:

Hypertrophic cardiomyopathy (HCM) is the most frequent genetic cardiac disorders and MYBPC3 and MYH7 are the most prevalent genes involved. MYBPC3Δ25bp carrier is a common variant and contributes to HCM through haploinsufficiency theory. A new variant with monoallelic double mutation MYBPC3Δ25bp/D389V is a subset of the common MYBPC3Δ25bp variant. The deletion of 25 bp variant affects MYBPC domains nearing C8-C10, while the D389V localizes at the C2 domain. In this manuscript, the C2 domain of the MYBPC with wild type or D389V were evaluated and compared through modeling systems. The findings showed that D389V had little effect on its interaction with myosin/actin protein, instead, caused conformational configuration shift and altered thermostability.

Data of modeling evaluation provided hypothesis for the explanation of pathomechanism for the development of HCM in MYBPC3Δ25bp/D389V variant. Minor suggestion was as follows.

  1. In the current version, the major and final conclusion was not found in the abstract. A new version is necessary.

We agree with the reviewer and rephrased parts of the abstract. In particular, we included the major and final conclusion the last sentence of the abstract: “Based on our data, we propose a pathomechanism for the development of HCM based on the toxic buildup of misfolded protein in young MYBPC3Δ25bp/D389V carriers that is supplanted and enhanced by C–zone haploinsufficiency at older ages.”

  1. MYBPC3Δ25bp is a common genetic variant for the development of HCM. MYBPPC3 D389V is a new variant found only in the subset of MYBPC3Δ25bp The D389V variant was not found alone. C2 domain with wild type or D389V was target for evaluation and comparison. It is curious to ask what findings in the presence of double mutations.

We thank the reviewer for pointing out the importance of studying the effects of the MYBPC3Δ25bp/D389V double mutation in both the C2 and C10 domains. Since these two domains differ greatly in terms of their localization and function in MyBPC, we judged it appropriate to study the effects of these mutations separately. The consequences of an altered C10 domain resulting from a potentially different splicing transcript have already been elucidated in vitro and in vivo [1–4]. However, the amount of MyBPC derived from the MYBPC3Δ25bp/D389V allele carrying a mutated C10 domain remains unclear, complicating research with the complete protein. Therefore, we focused our studies on the effects of the D389V mutation on the C2 domain of MyBPC and its N-terminal neighboring region. We conclude that the amount of MyBPC containing a mutated C10 domain correlates with the development of HCM in MYBPC3Δ25bp and MYBPC3Δ25bp/D389V carriers.

Reviewer 2 Report

In this study, the authors investigated the function of the Myosin–Binding Protein C Domain C2 and the effects of D389V mutation using several biochemical approaches, including molecular dynamics simulations, structural and termal stability assays. The work is relevant and well-written. The regulatory mechanisms of Myosin–Binding Protein C Domain C2 influenced by a single mutation were detailed. Minor suggestions are listed below.

  1. Has the degree of purity of the purified MyBPC protein been determined? How was it determined? The level of purity is a key factor for further assays and should be mentioned.
  2. Please identify the final numbering on the x and y axis from figure 5B.

Author Response

Reviewer #2:

In this study, the authors investigated the function of the Myosin–Binding Protein C Domain C2 and the effects of D389V mutation using several biochemical approaches, including molecular dynamics simulations, structural and termal stability assays. The work is relevant and well-written. The regulatory mechanisms of Myosin–Binding Protein C Domain C2 influenced by a single mutation were detailed. Minor suggestions are listed below.

  1. Has the degree of purity of the purified MyBPC protein been determined? How was it determined? The level of purity is a key factor for further assays and should be mentioned.

We thank the reviewer for the comment and agree that protein purity is a major requirement for obtaining reliable data from biochemical experiments. We included a Coomassie Brilliant Blue-stained SDS gel with all proteins used for the biochemical experiments in this study (Figure S2). The degree of purity of the prepared proteins was calculated by dividing the band intensity of the protein of interest by the total band intensity in a lane. The purity of β-cardiac myosin and actin amount to 95% and 98%, respectively. All MyBPC constructs have a purity of >99%.

  1. Please identify the final numbering on the x and y axis from figure 5B.

We thank the reviewer for the advice and updated Figure 5B, making it consistent with Figure 5A.

References

  1. Dhandapany, P.S.; Sadayappan, S.; Xue, Y.; Powell, G.T.; Rani, D.S.; Nallari, P.; Rai, T.S.; Khullar, M.; Soares, P.; Bahl, A.; et al. A Common MYBPC3 (Cardiac Myosin Binding Protein C) Variant Associated with Cardiomyopathies in South Asia. Nature genetics 2009, 41, 187–91, doi:10.1038/ng.309.
  2. Waldmüller, S.; Sakthivel, S.; Saadi, A.V.; Selignow, C.; Rakesh, P.G.; Golubenko, M.; Joseph, P.K.; Padmakumar, R.; Richard, P.; Schwartz, K.; et al. Novel Deletions in MYH7 and MYBPC3 Identified in Indian Families with Familial Hypertrophic Cardiomyopathy. Journal of Molecular and Cellular Cardiology 2003, 35, 623–636, doi:10.1016/S0022-2828(03)00050-6.
  3. Kuster, D.W.D.; Govindan, S.; Springer, T.I.; Martin, J.L.; Finley, N.L.; Sadayappan, S. A Hypertrophic Cardiomyopathy-Associated MYBPC3 Mutation Common in Populations of South Asian Descent Causes Contractile Dysfunction *. Journal of Biological Chemistry 2015, 290, 5855–5867, doi:10.1074/jbc.M114.607911.
  4. Kuster, D.W.D.; Lynch, T.L.; Barefield, D.Y.; Sivaguru, M.; Kuffel, G.; Zilliox, M.J.; Lee, K.H.; Craig, R.; Namakkal-Soorappan, R.; Sadayappan, S. Altered C10 Domain in Cardiac Myosin Binding Protein-C Results in Hypertrophic Cardiomyopathy. Cardiovascular Research 2019, doi:10.1093/cvr/cvz111.

Round 2

Reviewer 1 Report

There is no addition comment.